# Reliability and validity of the Forgotten Joint Score-12 for total ankle replacement and ankle arthrodesis

Koji Noguchi[1,2], Satoshi Yamaguchi[3,4]*, Atsushi Teramoto[5], Kentaro Amaha[6], Noriyuki Kanzaki[7], Hirofumi Tanaka[8], Tetsuro Yasui[9], Yosuke Inaba[10]

1 Department of Orthopaedic Surgery, Kurume University School of Medicine, Kurume-shi, Fukuoka, Japan, 2 Japan Community Health care Organization Kurume General Hospital, Kurume-shi, Fukuoka, Japan, 3 Graduate School of Global and Transdisciplinary Studies, Chiba University, Chiba-shi, Chiba, Japan, 4 Graduate School of Medical and Pharmaceutical Sciences, Chiba University, Chiba-shi, Chiba, Japan, 5 Department of Orthopaedic Surgery, Sapporo Medical University School of Medicine, Sapporo-shi, Hokkaido, Japan, 6 Department of Orthopaedic Surgery, St. Luke's International Hospital, Chuo-ku, Tokyo, Japan, 7 Department of Orthopaedic Surgery, Kobe University Graduate School of Medicine, Kobe-shi, Hyogo, Japan, 8 Hyakutake Orthopedic Surgery and Sports Clinic, Saga-shi, Saga, Japan, 9 Department of Orthopaedic Surgery, Teikyo University Mizonokuchi Hospital, Kawasaki-shi, Kanagawa, Japan, 10 Biostatistics Section, Chiba University Hospital Clinical Research Center, Chiba-shi, Chiba, Japan

* y-satoshi@faculty.chiba-u.jp

**Data Availability Statement:** All relevant data are within the paper and its Supporting information files.

## Abstract

### Objectives

This study evaluated the reliability and validity of the Forgotten Joint Score-12 (FJS-12)—a measure of patients' ability to forget their joints in daily life—in patients who underwent total ankle replacement (TAR) or ankle arthrodesis (AA).

### Methods

Patients who underwent TAR or AA were recruited from seven hospitals. The patients completed the Japanese version of FJS-12 twice, at an interval of two weeks, at a minimum of one year postoperatively. Additionally, they answered the Self-Administered Foot Evaluation Questionnaire and EuroQoL 5-Dimension 5-Level as comparators. The construct validity, internal consistency, test-retest reliability, measurement error, and floor and ceiling effects were evaluated.

### Results

A total of 115 patients (median age, 72 years), comprising 50 and 65 patients in the TAR and AA groups respectively, were evaluated. The mean FJS-12 scores were 65 and 58 for the TAR and AA groups, respectively, with no significant difference between groups ($P$ = 0.20). Correlations between the FJS-12 and Self-Administered Foot Evaluation Questionnaire subscale scores were good to moderate. The correlation coefficient ranged from 0.39 to 0.71 and 0.55 to 0.79 in the TAR and AA groups, respectively. The correlation between the FJS-12 and EuroQoL 5-Dimension 5-Level scores was poor in both groups. The internal consistency was adequate, with Cronbach's α greater than 0.9 in both groups. The

**Funding:** The author(s) received no specific funding for this work.

**Competing interests:** The authors have declared that no competing interests exist.

intraclass correlation coefficients of test-retest reliability was 0.77 and 0.98 in the TAR and AA groups, respectively. The 95% minimal detectable change values were 18.0 and 7.2 points in the TAR and AA groups, respectively. No floor or ceiling effect was observed in either group.

## Conclusions

The Japanese version of FJS-12 is a valid and reliable questionnaire for measuring joint awareness in patients with TAR or AA. The FJS-12 can be a useful tool for the postoperative assessment of patients with end-stage ankle arthritis.

## Introduction

Total ankle replacement (TAR) and ankle arthrodesis (AA) are two common surgical procedures used to treat end-stage ankle arthritis. Advancements in operative techniques and aging population has increased the number of these operations worldwide [1, 2].

The significance of patient-reported outcome measures (PROMs) is increasingly recognized for the postoperative evaluation of TAR and AA. Various PROMs have been used to measure foot and ankle outcomes [3]. However, no single outcome accurately depicts the difficulties that patients with end-stage ankle arthritis [4]. Additionally, some foot and ankle instruments have insufficient psychometric properties such as floor and ceiling effects [5] and limited validity [6]. Consequently, there is no consensus on the most appropriate instrument for this patient population [7]. Furthermore, several existing PROMs are lengthy, which burdens patients and may lead to loss of follow-up [8]. A simple and sufficiently validated PROM is therefore required.

The Forgotten Joint Score-12 (FJS-12) is a new PROM based on the assumption that the ultimate goal of a joint operation is to forget the joint [9]. The FJS-12 consists of 12 questions on joint awareness of daily activities, such as walking, climbing stairs, and doing household chores. It was originally introduced for postoperative assessment of total knee or hip arthroplasty, and its validity and reliability have been documented [10]. This measurement had no or low floor and ceiling effects, indicating that it could distinguish between patients with good and excellent outcomes [10]. Consequently, FJS-12 has been translated into different languages and is increasingly used worldwide. The Japanese version was created using a cross-cultural translation and adaptation process (S1 File) [11]. Its reliability and validity have been tested in patients who have undergone total knee and hip arthroplasty [11].

The FJS-12 has recently been used in other groups of patients, such as those who underwent unicompartmental knee arthroplasty, high tibial osteotomy, and patellar dislocation [12, 13]. The questions in the FJS-12 can be applied to the assessment of patients with foot and ankle diseases because the questions include items on symptoms during locomotive activities. Although the usefulness of FJS-12 has not been proven, it was used in a study to evaluate patients with hallux rigidus [14]. To date, the psychometric properties of the FJS-12, either the original English or Japanese version, have not been tested in patients who have undergone TAR and AA. Therefore, this study aimed to evaluate the reliability and validity of the Japanese version of the FJS-12 for the TAR and AA. We hypothesized that the Japanese version of the FJS-12 would be valid and reliable to measure joint awareness after TAR or AA.

## Materials and methods

### Patient recruitment

Consecutive patients were recruited from the foot and ankle outpatient clinics of seven hospitals between February and November 2021. Patients who underwent TAR or AA and had a minimum follow-up duration of one year were included in this study. Patients who were unable to answer the questionnaires owing to cognitive impairment or mental disorders, those who were unable to walk independently at the time of the questionnaire survey, or those with a history of neurological disorders, such as diabetic neuropathy, were excluded. Patients who did not provide consent to participate were excluded.

The research ethics committees of the institutions (Kurume University School of Medicine, approval number, 20239; Graduate School of Medicine, Chiba University, approval number, 4052; Hyakutake Orthopedic Surgery and Sports Clinic, approval number, 2102; St. Luke's International Hospital, approval number, 21-R007; Sapporo Medical University School of Medicine, approval number, 332–9; Kobe University Graduate School of Medicine, approval number, B210057; Teikyo University Mizonokuchi Hospital, approval number, TUIC-COI21-0150) approved this study, and all patients provided written informed consent.

### Patients' demographics

The patients' demographics, including the type of operative procedure (TAR or AA), age, sex, diagnosis, postoperative follow-up duration, laterality of the operation (unilateral or bilateral), and primary/revision surgery, were obtained from their medical records. The implant type was recorded for patients who underwent TAR. Regarding patients who underwent AA, details of the operative procedure (arthroscopic or open) were surveyed.

### Questionnaires

The patients answered the questionnaires in an outpatient setting for more than one year postoperatively. In the first round of evaluation, they completed the FJS-12 [9], Self-Administered Foot Evaluation Questionnaire (SAFE-Q) [15], and EuroQoL 5-Dimension 5-Level (EQ-5D-5L) [16]. In the second round of evaluation (two weeks after the first evaluation), the patients were asked to complete the FJS-12 and global rating scale. Patients who underwent bilateral ankle surgery answered a questionnaire based on the side that was first operated on.

The FJS-12 consists of 12 questions on the respondent's joint awareness related to activities of daily living [9]. Patients assessed each item on a 5-point Likert scale: never (0), almost never (1), seldom (2), sometimes (3), and mostly (4). The item scores were summed and converted into a scale ranging from 0 to 100, with a higher score indicating a patient's greater ability to forget the affected joint. Patients with more than four missing answers were excluded from the analysis [9].

The SAFE-Q is a 34-item questionnaire that has proven to be a good tool for assessing the quality of life of patients with foot and ankle diseases [15]. The SAFE-Q has been widely used for evaluating patients with end-stage ankle arthritis [17, 18]. The outcome consists of five subscales: pain and pain-related, physical functioning and daily living, social functioning, shoe-related, and general health and well-being. Responses to each question were rated from 0 to 4 and then converted to subscale scores ranging from 0 to 100, with a higher score indicating a better ankle condition.

The EQ-5D-5L is a self-reported measure used to evaluate general health status [16]. It consists of the descriptive system and EQ visual analog scale (VAS). The descriptive system comprises five dimensions: mobility, self-care, usual activities, pain/discomfort, and anxiety/

depression. The answers to each dimension were in five levels, ranging from no problems (1) to extreme problems (5). The responses to the five dimensions were converted into a single summary number called an index value. The possible score ranged from -0.025 to 1, and a value of 1 indicated full health. The EQ VAS records the patient's self-rated health on a vertical scale, where the end-points are labeled "the best health you can imagine (100)" and "the worst health you can imagine (0)."

The global rating scale was used to examine whether the global condition of the operated ankle changed between the first and second evaluations. The answers were rated from (1) completely recovered to (7) vastly worsened [19]. Based on a previous report, patients rated between 3 and 5 in the second evaluation were considered stable and were included in the analysis [19]. Patients rated 1, 2, 6, or 7 were excluded because their condition changed between evaluations [19].

## Statistics

The patients were grouped into two: those who had undergone TAR (TAR group) and AA (AA group). Patients' demographics and questionnaire results were compared between the TAR and AA groups using the Wilcoxon signed-rank tests, Student t-test, and Fisher's exact test as appropriate.

**Distribution of FJS-12 scores.** The normality of the FJS-12 scores was examined using the Shapiro–Wilk test. The percentage of missing responses was then calculated.

**Construct validity and hypothesis testing.** Regarding construct validity, correlations between the FJS-12 and subscale scores of the SAFE-Q and EQ-5D were assessed using Spearman's correlation coefficients. Correlation coefficients higher than 0.7 were considered good, 0.3 to 0.7 moderate, and lower than 0.3 poor [20]. We hypothesized that the correlation between the FJS-12 and SAFE-Q subscales would be good to moderate. We also hypothesized that the correlation between the FJS-12 and EQ-5D-5L would be moderate to poor because the EQ-5D score is affected by factors such as comorbidities that are unrelated to the ankle joint [21]. To assess known-group validity, the FJS-12 scores were compared between the TAR and AA groups using the Student t-test. We hypothesized that the scores were similar between the groups. If at least 75% of the hypotheses (12 out of 15 hypotheses) were confirmed, the construct validity was considered adequate [22].

**Internal consistency.** Internal consistency was assessed using Cronbach's α, which was considered adequate if the value was between 0.70 and 0.95 [22].

**Factor analysis.** Principal component analysis with Varimax rotation was performed to assess the dimensionality of the FJS-12 using the overall data of the TAR and AA groups.

**Test-retest reliability.** The test-retest reliability was evaluated by calculating the intraclass correlation coefficient $(ICC)_{agreement}$ for a two-way random-effects model, single measurement. The ICC was considered adequate if the value was higher than 0.70 [22].

**Measurement error.** The standard error of measurement (SEM) and 95% minimal detectable change ($MDC^{95}$) were used to express measurement error. SEM was calculated as follows: $SEM = SD \times \sqrt{(1 - ICC)}$, where SD is the standard deviation of the difference between the two scores. $MDC^{95}$ was calculated as follows: $MDC^{95} = 1.96 \times \sqrt{2} \times SEM$ [22].

**Floor and ceiling effects.** Floor and ceiling effects for the FJS-12 were considered present if more than 15% of the patients had the lowest (0) or highest (100) scores [22].

**Sample size.** The sample size was determined based on the recommendation of the COnsensus-based Standards for the selection of health Measurement INstruments, in which 50 or more patients were adequate in assessing the measurement property of a PROM [23]. Assuming that 20% of patients were excluded from the analysis, we determined a sample size of 63

patients for each group (50/0.8 = 62.5). Statistical analyses were performed using commercially available statistical software (JMP Pro 15.1.0, SAS Institute, Cary, NC, USA and Bell Curve for Excel, Social Survey Research Information, Shinjuku, Tokyo, Japan). Statistical significance was set at a *P*-value less than 0.05.

## Results

### Patients' demographics

Among the 142 patients screened for eligibility, 27 were excluded. Data from the remaining 115 patients, 50 in the TAR group (Fig 1A) and 65 in the AA group (Fig 1B), were analyzed. There were 87 women and 28 men, with a median age of 72 (25th and 75th percentiles: 67 and 79) years (Table 1).

### Distribution of FJS-12 scores

The FJS-12 scores were normally distributed in both TAR (*P* = 0.48, Fig 2A) and AA groups (*P* = 0.14, Fig 2B). The overall proportion of missing responses was 28/600 (5%) and 12/780 (2%) in the TAR and AA groups, respectively. Question 12, an item on sports activities, had the highest percentage of missing responses with 20/50 (40%) and 7/65 (11%) missing responses in the TAR and AA groups, respectively.

### Construct validity

In the TAR group, the correlations between the FJS-12 and SAFE-Q subscale scores were good to moderate, with the correlation coefficient ranging from 0.39 to 0.71 (Table 2). There was a moderate correlation between the FJS-12 and EQ-5D-5L descriptive system index values, but no correlation between the FJS-12 and EQ VAS scores (Table 2). In the AA group, the correlation coefficient between the FJS-12 and SAFE-Q subscales ranged from 0.55 to 0.79 (Table 2). The correlations between the FJS-12 and EQ-5D-5L were also moderate to good. There was no significant difference in the FJS-12 score between the two groups, although the score was 7 points higher in the TAR group than in the AA group (*P* = 0.20, Table 1). Among the 15 hypotheses, 14 corresponded with our hypotheses.

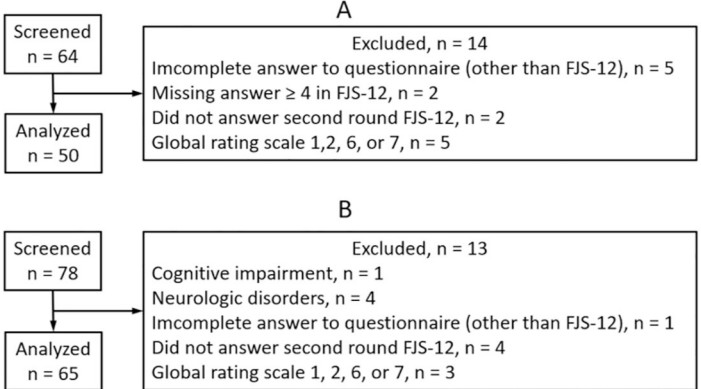

**Fig 1. Patient flow diagram.** (A) patient after total ankle replacement, (B) patients after ankle arthrodesis.

**Table 1. Patient demographics and outcome scores.**

| | | TAR group (n = 50) | AA group (n = 65) | *P* |
|---|---|---|---|---|
| Age (years) | | 78 (72, 81) | 70 (62, 73) | < 0.001[a] |
| Sex (Women, n [%]) | | 40 (80) | 47 (72) | 0.39[b] |
| Diagnosis (n [%]) | Idiopathic OA | 23 (46) | 24 (37) | 0.008[b] |
| | Post-traumatic OA | 8 (16) | 13 (20) | |
| | Post-sprain OA | 7 (14) | 24 (37) | |
| | Rheumatoid arthritis | 9 (18) | 3 (5) | |
| | Others | 3 (6) | 1 (2) | |
| Postoperative follow-up (months) | | 29 (17, 56) | 34 (19, 51) | 0.48[a] |
| Operative procedure (n [%]) | TNK ankle | 13 (26) | n.a. | |
| | TNK ankle + total talar prosthesis | 17 (34) | n.a. | |
| | FINE Ankle | 1 (2) | n.a. | |
| | TM Ankle | 19 (38) | n.a. | |
| | Arthroscopic arthrodesis | n.a. | 60 (92) | |
| | Open arthrodesis | n.a. | 5 (8) | |
| Bilateral operation (n [%]) | | 2 (4) | 3 (5) | 1.00[b] |
| Revision operation (n [%]) | | 1 (2) | 3 (5) | 0.63[b] |
| Interval between 1st and 2nd FJS (days) | | 14 (14, 14) | 14 (14, 15) | 0.09[a] |
| FJS-12 (mean ± standard deviation) | | 64 ± 20 | 58 ± 26 | 0.20[c] |
| SAFE-Q | Pain | 87 (77, 100) | 94 (79, 100) | 0.15[a] |
| | Physical Functioning | 76 (68, 87) | 77 (64, 89) | 0.79[a] |
| | Social Functioning | 88 (74, 100) | 100 (75, 100) | 0.06[a] |
| | Shoe-Related | 83 (67, 100) | 83 (67, 96) | 0.27[a] |
| | General Health | 93 (80, 100) | 90 (80, 100) | 0.94[a] |
| EQ-5D-5L | Descriptive system index value | 0.86 (0.74, 0.89) | 0.89 (0.78, 1) | 0.21[a] |
| | EQ VAS | 80 (74, 90) | 80 (70, 90) | 0.63[a] |

Values indicate the median (25th and 75th percentiles) unless indicated otherwise. TAR, total ankle replacement; AA, ankle arthrodesis; FJS, Forgotten Joint Score; SAFE-Q, Self-Administered Foot Evaluation Questionnaire; EQ-5D-5L, EuroQoL 5-dimension 5-level; VAS, visual analogue scale; OA, osteoarthritis; TNK Ankle (Kyocera, Fushimi-cho, Kyoto, Japan); FINE Ankle (Teijin Nakashima Medical, Higashi-ku, Okayama, Japan); TM Ankle (Zimmer, Biomet, Warsaw, IN, USA); n.a., not applicable; n; number of patients.

[a]Wilcoxon signed-rank test.

[b]Fisher's exact test.

[c]Student t-test.

## Internal consistency

A high level of internal consistency was found, with Cronbach's α of 0.92 and 0.94 in the TAR and AA groups, respectively.

## Factor analysis

The first factor explained 58.9% of the variance, with an eigenvalue of 7.06 (Fig 3). The first factor mainly represented difficulties during activities, such as items 3, 6–8, 11, and 12 (Table 3). The second factor explained 9.9% of the variance, with an eigenvalue of 1.19. This factor mainly represented difficulties at rest, such as items 1, 2, 4, and 5.

## Test-retest reliability and measurement error

In the TAR group, the ICC of the FJS-12 score was 0.77 (95% confidence interval [CI], 0.63 to 0.86), showing adequate reliability; the SEM was 6.5, and the MDC[95] was 18.0 (S2 File). The

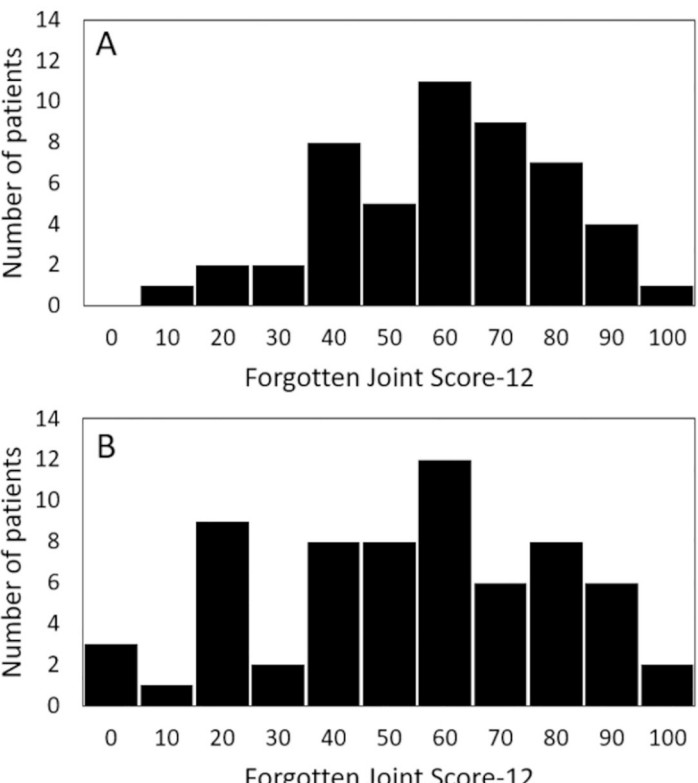

**Fig 2. Distribution of FJS-12 scores.** (A) patient after total ankle replacement, (B) patients after ankle arthrodesis.

reliability in the AA group was also sufficient, with an ICC of 0.93 (95% CI, 0.89 to 0.96); the SEM value was 2.6, and the MDC$^{95}$ value was 7.2 (S2 File).

## Floor and ceiling effects

None of the patients in the TAR group had a score of 0 and one patient (2%) had a score of 100. In the AA group, three (5%) patients scored 0, and two (3%) patients scored 100 (Fig 2). Therefore, there was no floor or ceiling effect in either group.

**Table 2. Correlations of the FJS-12 with the SAFE-Q and EQ-5D-5L.**

| | | TAR group (n = 50) | | AA group (n = 65) | |
|---|---|---|---|---|---|
| | | ρ | P | ρ | P |
| SAFE-Q | Pain | 0.71 | < 0.001 | 0.62 | < 0.001 |
| | Physical functioning | 0.61 | < 0.001 | 0.79 | < 0.001 |
| | Social functioning | 0.39 | 0.006 | 0.71 | < 0.001 |
| | Shoe-related | 0.61 | < 0.001 | 0.55 | < 0.001 |
| | General health | 0.59 | < 0.001 | 0.68 | < 0.001 |
| EQ-5D-5L | Descriptive system index value | 0.34 | 0.01 | 0.70 | < 0.001 |
| | EQ VAS | 0.05 | 0.73 | 0.51 | < 0.001 |

TAR, total ankle replacement; AA, ankle arthrodesis; FJS, Forgotten Joint Score; SAFE-Q, Self-Administered Foot Evaluation Questionnaire; EQ-5D-5L, EuroQoL 5-dimension 5-level; VAS, visual analog scale; ρ, Spearman correlation coefficient; n; number of patients.

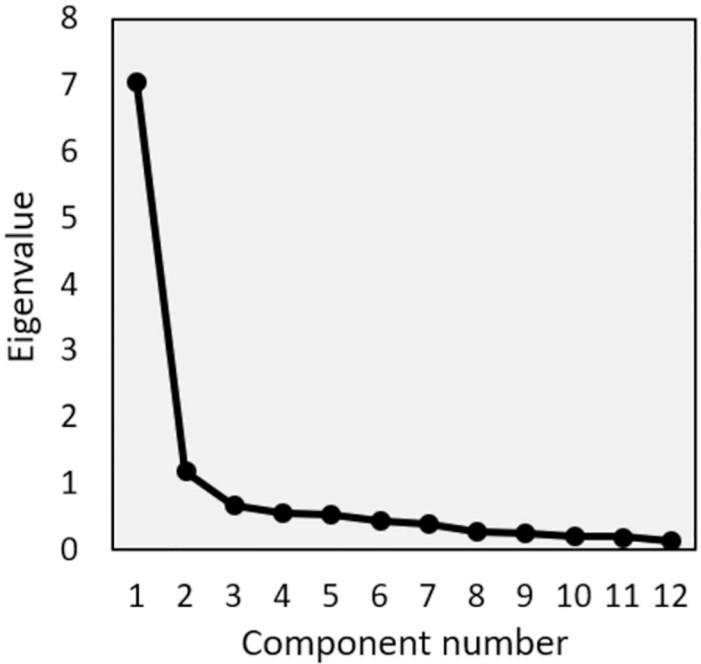

**Fig 3. Scree plot of FJS-12.**

## Discussion

This study showed that FJS-12 scores were moderately to highly correlated with SAFE-Q scores after operative treatment for end-stage ankle arthritis. The test-retest reliability and measurement errors were acceptable. No floor or ceiling effects were observed.

Our data showed good-to-moderate correlations between the FJS-12 and SAFE-Q subscale scores in both the TAR and AA groups. The correlation coefficients between FJS-12 and EQ-5D-5L scores were lower than those between FJS-12 and SAFE-Q scores. A good correlation

**Table 3. Factor loading of the FJS-12.**

| FJS 12 items | Factor 1 | Factor 2 |
|---|---|---|
| 1 | 0.249 | **0.680** |
| 2 | 0.237 | **0.872** |
| 3 | **0.599** | **0.521** |
| 4 | **0.418** | **0.604** |
| 5 | 0.385 | **0.622** |
| 6 | **0.736** | 0.212 |
| 7 | **0.830** | 0.295 |
| 8 | **0.735** | 0.294 |
| 9 | **0.733** | 0.348 |
| 10 | **0.665** | 0.329 |
| 11 | **0.720** | **0.457** |
| 12 | **0.662** | 0.350 |

Bold numbers indicate the factor loading > 0.4.

between the FJS-12 and SAFE-Q scores is reasonable because they include similar questions, such as items about walking for a long time, walking on uneven ground, and climbing stairs [9, 15]. The FJS-12 has been shown to have good construct validity in evaluating patients after total knee and hip arthroplasty, high tibial osteotomy, anterior cruciate ligament reconstruction, or hip arthroscopy [10, 24–26]. The FJS-12 may be used as a simple and valid instrument for assessing TAR and AA.

There was no significant difference in the FJS-12 score between the TAR and AA groups, although the score was slightly higher in the TAR group than in the AA group. This study, therefore, confirmed the construct validity of the questionnaire. Our results were consistent with a recent systematic review that found equality in PROMs after TAR and AA [27]. However, the patients in the TAR group were 8 years older than those in the AA group. Although not the main objective of this study, a comparison of TAA and AA requires further research.

In this study, the principal component analysis showed the two dimensions of the FJS-12, although the eigenvalue for the first factor was six times larger than that of the second factor (7.06 versus 1.09). In contrast, studies using patients after total knee arthroplasty and anterior cruciate reconstruction reported unidimensionality of the FJS-12 [24, 28]. The differences in the patient background and analysis methods are possible explanations for the discrepancy. Further studies are required to confirm the factor structure of the FJS-12 for patients after TAR and AA.

In this study, the test-retest reliability was adequate for patients who underwent AA. In patients who underwent TAR, the ICC was 0.77; however, it was higher than the minimum standard for reliability [22]. A recent systematic review found that the test-retest reliability of the FJS-12 was higher than 0.8 in patients who underwent total knee and hip arthroplasties [10]. A more recent study reported an ICC value of 0.76 after total joint arthroplasty [29]. Therefore, the FJS-12 was as reliable for patients with end-stage ankle arthritis after surgery as it was for those with other joint diseases.

This study showed that the MDC$^{95}$ values for the Japanese FJS-12 were 18.0 and 7.2 points in the TAR and AA groups, respectively. The results for the MDC$^{95}$ are comparable to those obtained for patients treated with total knee arthroplasty, which ranged from 13 to 24 points [30, 31]. Similarly, the MDC$^{95}$ was between 14 and 21 points after total hip arthroplasty [30, 32]. However, the patients who underwent hip arthroscopy had a high MDC$^{95}$ of 32 points [33]. The clinical explanation for this variability is that patients who underwent AA had stable levels of postoperative joint awareness, perhaps because of the loss of ankle mobility. Conversely, patients who underwent TAR and total knee arthroplasties experienced fluctuations in symptoms, which would reasonably be expected for reconstructed mobile joints.

The FJS-12 had no ceiling or floor effects in patients who underwent TAR or AA. This contrasts with the SAFE-Q, in which most subscales had ceiling effects. The results of this study suggest that a forgotten ankle is difficult to achieve, even after successful TAR and AA. Similar to the findings of our study, the FJS-12 had little or no ceiling effect in total knee and hip arthroplasties [10]. The results also indicate that the FJS-12 effectively detects small differences among well-functioning patients. The evaluation of these patients after TAR will become more important because operative techniques and implant designs have advanced over the past decade, leading to better outcomes. However, further research is warranted to determine whether small differences in the FJS-12 scores are clinically relevant.

This study has several limitations. First, the patients were relatively old, possibly owing to Japan's aging population. Further studies with a wider age range are required. Second, the responsiveness of the FJS-12 score was not assessed owing to the cross-sectional nature of this study. Longitudinal studies are required to clarify whether FJS-12 can capture changes in patient status over time. Third, the content validity of the FJS-12 was not assessed in the

population with ankle disease; consequently, the items contained in the questionnaire may not be comprehensive or relevant for this patient population. Fourth, the SAFE-Q was used to assess the ankle disease-specific outcomes. This is because other commonly used instruments, such as the European Foot and Ankle Society Score and Foot and Ankle Outcome Score [34, 35], do not have a Japanese version. However, the SAFE-Q has been widely used to evaluate patients treated with TAR and AA [17, 18]. Fifth, we used the Japanese version of the FJS-12 [11]. The validity and reliability of other language versions of the FJS-12 require confirmation in future studies. Finally, the number of patients was not based on the sample size calculation. However, we determined the sample size according to the recommendation of the widely accepted guidelines [23]. Furthermore, the post hoc statistical power analysis using the correlation between the FJS-12 and SAFE-Q scores showed the statistical power ranged from 0.80 to 0.99.

In conclusion, the Japanese version of the FJS-12 was a valid and reliable PROM that could differentiate between patients with excellent and good outcomes after the operative treatment of end-stage ankle arthritis. The FJS-12 can be a useful tool for the postoperative assessment of patients with end-stage ankle arthritis.

## Supporting information

**S1 File. Japanese version of FJS-12.**
(DOCX)

**S2 File. FJS scores.**
(XLSX)

## Author Contributions

**Conceptualization:** Koji Noguchi, Satoshi Yamaguchi.

**Data curation:** Koji Noguchi, Satoshi Yamaguchi, Atsushi Teramoto, Kentaro Amaha, Noriyuki Kanzaki, Hirofumi Tanaka, Tetsuro Yasui.

**Formal analysis:** Koji Noguchi, Atsushi Teramoto, Kentaro Amaha, Yosuke Inaba.

**Investigation:** Koji Noguchi.

**Methodology:** Satoshi Yamaguchi, Atsushi Teramoto, Kentaro Amaha, Yosuke Inaba.

**Project administration:** Satoshi Yamaguchi.

**Supervision:** Tetsuro Yasui.

**Validation:** Noriyuki Kanzaki, Hirofumi Tanaka, Yosuke Inaba.

**Writing – original draft:** Koji Noguchi, Satoshi Yamaguchi.

**Writing – review & editing:** Atsushi Teramoto, Kentaro Amaha, Noriyuki Kanzaki, Hirofumi Tanaka, Tetsuro Yasui, Yosuke Inaba.

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
