## [Decision Letter · Decision Letter 0]

24 Mar 2023

PONE-D-23-04534Reliability and validity of the Forgotten Joint Score-12 for total ankle replacement and ankle arthrodesisPLOS ONE

Dear Dr. Yamaguchi,

Thank you for submitting your manuscript to PLOS ONE. After careful consideration, we feel that it has merit but does not fully meet PLOS ONE’s publication criteria as it currently stands. Therefore, we invite you to submit a revised version of the manuscript that addresses the points raised during the review process.

We look forward to receiving your revised manuscript.

Kind regards,

Fatih Özden, PhD

Academic Editor

PLOS ONE

Journal Requirements:

Additional Editor Comments:

Please find the reviewer reports below. Kind Regards

Reviewers' comments:

Reviewer's Responses to Questions

**Comments to the Author**

1. Is the manuscript technically sound, and do the data support the conclusions?

Reviewer #1: No

Reviewer #2: Yes

Reviewer #3: Yes

2. Has the statistical analysis been performed appropriately and rigorously? 

Reviewer #1: Yes

Reviewer #2: Yes

Reviewer #3: I Don't Know

3. Have the authors made all data underlying the findings in their manuscript fully available?

Reviewer #1: Yes

Reviewer #2: Yes

Reviewer #3: Yes

4. Is the manuscript presented in an intelligible fashion and written in standard English?

Reviewer #1: Yes

Reviewer #2: Yes

Reviewer #3: No

5. Review Comments to the Author

Reviewer #1: Dear authors,

this is an interesting study and should have been conducted earlier, this will help the F&A society to improve outcome measurements. I have the following comments:

- English could be shortened

- Spelling mistakes on page 12 line 201 and 202

-Why did you not statistic testing on patients groups (Tbl1)

These are minor issues and the excellent study is worth publishing in my view.

Best regards

Reviewer #2: The manuscript aimed to examine the reliability and validity of the Japanese Forgotten Joint Score-12 (FJS-12) in patients with total ankle replacement (TAR) and ankle arthrodesis (AA). The manuscript is interesting, well-written, and presented in an organized fashion. I have only one comment on the work as presented below:

1- Abstract, please report the SEM and MDC in the Results section. As well, the Conclusion should consider that the Japanese version has been evaluated for metric characteristics and found reliable and valid.

2- Please use appropriate statistical test to compare the two groups of TAR and AA on the FJS-12 scores. The result may indicate the discriminant ability of the questionnaire.

Reviewer #3: The authors have provided a purposeful and unique study. A comprhensive English editing is required to shorten the length of the sentences and also grammatical issues should be handled to improve the comprehensibility of the paper. Alsa, formula based sample size calculation is required.

6. PLOS authors have the option to publish the peer review history of their article (what does this mean?). If published, this will include your full peer review and any attached files.

Reviewer #1: No

Reviewer #2: **Yes: **Noureddin Nakhostin Ansari

Reviewer #3: No

---

## [Author Response · Author response to Decision Letter 0]

4 Apr 2023

Response to Reviewers file was attached to the revised manuscript.

---

## [Decision Letter · Decision Letter 1]

24 Apr 2023

PONE-D-23-04534R1Reliability and validity of the Forgotten Joint Score-12 for total ankle replacement and ankle arthrodesisPLOS ONE

Dear Dr. Yamaguchi,

Thank you for submitting your manuscript to PLOS ONE. After careful consideration, we feel that it has merit but does not fully meet PLOS ONE’s publication criteria as it currently stands. Therefore, we invite you to submit a revised version of the manuscript that addresses the points raised during the review process.

Additional Editor Comments:

Dear Authors,

Please find one of the reviewers' comments, both major and minor issues raised. I look forward your corrected paper soon.

Best Regards

We look forward to receiving your revised manuscript.

Kind regards,

Fatih Özden, PhD

Academic Editor

PLOS ONE

Journal Requirements:

Reviewers' comments:

Reviewer's Responses to Questions

**Comments to the Author**

1. If the authors have adequately addressed your comments raised in a previous round of review and you feel that this manuscript is now acceptable for publication, you may indicate that here to bypass the “Comments to the Author” section, enter your conflict of interest statement in the “Confidential to Editor” section, and submit your "Accept" recommendation.

Reviewer #2: (No Response)

Reviewer #3: All comments have been addressed

2. Is the manuscript technically sound, and do the data support the conclusions?

Reviewer #2: Yes

Reviewer #3: Yes

3. Has the statistical analysis been performed appropriately and rigorously? 

Reviewer #2: Yes

Reviewer #3: Yes

4. Have the authors made all data underlying the findings in their manuscript fully available?

Reviewer #2: Yes

Reviewer #3: Yes

5. Is the manuscript presented in an intelligible fashion and written in standard English?

Reviewer #2: Yes

Reviewer #3: Yes

6. Review Comments to the Author

Reviewer #2: I appreciate the efforts of the authors in revising the manuscript. The manuscript looks better, but there are still issues that is required to be addressed.

Abstract

1- Results, please report the results on know-group validity.

2- Conclusions, lines 55-57 are repetitions with results. I would suggest omitting them.

Introduction

1- Page 6, line 87, "I would suggest omitting "effectiveness" and inserting "usefulness", please.

Metods

1- Page 10, line 171, please clarify the statement on the confirmation of 75% of hypotheses for construct validity in the context of your study.

2- Page 10, line 172, please analyze the exploratory factor analysis considering all the data on Japanese FJS-12 from both groups (n==115) .

3- Page 10, line 177, on ICC, Please clarify, Single measure or average measure? Further, absolute or consistency? ICC agreement was used? Please also provide reason/reasons for using the two-way mixed-effects model.

4- Page 11, line 191, how 63 patients were calculatedfor sample size? Terwee et al (2007) suggest at least 100 patients. This needs clarification.

Results

1- Page 12, Table 1, last coloumn, please report the tests used along with p values.

2- Page 17, lines 253-259, you were needed to analyze the floor and ceiling effects for FJS-12 scores. You may omit reports findings on the floor and ceiling effects for SAFE-Q scores and EQ-5D-5L. Please clarify it.

Discussion

1- This section should also focus on the examination of the Japanese FJS-12 scores known-group validity that was not confirmed. Findings on the factor analysis should also considered for the revised paper.

2- Page 20, Conclusions, please avoid repetitions of the results and focus instead on the outcomes of the study, and whether you were successful in addressing the gaps as already stated in the Introduction.

Reviewer #3: Thanks for your revision. The manuscript is now suitable for publication regarding my point of view.

7. PLOS authors have the option to publish the peer review history of their article (what does this mean?). If published, this will include your full peer review and any attached files.

Reviewer #2: **Yes: **Noureddin Nakhostin Ansari

Reviewer #3: No

---

## [Author Response · Author response to Decision Letter 1]

20 May 2023

Responses were attached to the submitted file

---

## [Editor Report · Decision Letter 2]

23 May 2023

Reliability and validity of the Forgotten Joint Score-12 for total ankle replacement and ankle arthrodesis

PONE-D-23-04534R2

Dear Dr. Yamaguchi,

We’re pleased to inform you that your manuscript has been judged scientifically suitable for publication and will be formally accepted for publication once it meets all outstanding technical requirements.

Kind regards,

Fatih Özden, PhD

Academic Editor

PLOS ONE
---

## [Editor Report · Acceptance letter]

5 Jun 2023

PONE-D-23-04534R2 

Reliability and validity of the Forgotten Joint Score-12 for total ankle replacement and ankle arthrodesis 

Dear Dr. Yamaguchi:

I'm pleased to inform you that your manuscript has been deemed suitable for publication in PLOS ONE. Congratulations! Your manuscript is now with our production department. 

Kind regards, 

on behalf of

Dr. Fatih Özden 

Academic Editor

PLOS ONE